# A 4-Hydroxybenzoic Acid-Mediated Signaling System Controls the Physiology and Virulence of *Shigella sonnei*

Mingfang Wang,[a] Jia Zeng,[a] Yu Zhu,[a] Xiayu Chen,[a] Quan Guo,[a] Huihui Tan,[a] Binbin Cui,[a] Shihao Song,[a,b] Yinyue Deng[a,b]

[a]School of Pharmaceutical Sciences (Shenzhen), Sun Yat-sen University, Shenzhen, China
[b]School of Pharmaceutical Sciences, Hainan University, Haikou, China

**ABSTRACT** Many bacteria use small molecules, such as quorum sensing (QS) signals, to perform intraspecies signaling and interspecies or interkingdom communication. Previous studies demonstrated that some bacteria regulate their physiology and pathogenicity by employing 4-hydroxybenzoic acid (4-HBA). Here, we report that 4-HBA controls biological functions, virulence, and anthranilic acid production in *Shigella sonnei*. The biosynthesis of 4-HBA is performed by UbiC (SSON_4219), which is a chorismate pyruvate-lyase that catalyzes the conversion of chorismate to 4-HBA. Deletion of *ubiC* caused *S. sonnei* to exhibit impaired phenotypes, including impaired biofilm formation, extracellular polysaccharide (EPS) production, and virulence. In addition, we found that 4-HBA controls the physiology and virulence of *S. sonnei* through the response regulator AaeR (SSON_3385), which contains a helix-turn-helix (HTH) domain and a LysR substrate-binding (LysR_substrate) domain. The same biological functions are controlled by AaeR and the 4-HBA signal, and 4-HBA-deficient mutant phenotypes were rescued by in *trans* expression of AaeR. We found that 4-HBA binds to AaeR and then enhances the binding of AaeR to the promoter DNA regions in target genes. Moreover, we revealed that 4-HBA from *S. sonnei* reduces the competitive fitness of *Candida albicans* by interfering with morphological transition. Together, our results suggested that the 4-HBA signaling system plays crucial roles in bacterial physiology and interkingdom communication.

**IMPORTANCE** *Shigella sonnei* is an important pathogen in human intestines. Following previous findings that some bacteria employ 4-HBA as a QS signal to regulate biological functions, we demonstrate that 4-HBA controls the physiology and virulence of *S. sonnei*. This study is significant because it identifies both the signal synthase UbiC and receptor AaeR and unveils the signaling pathway of 4-HBA in *S. sonnei*. In addition, this study also supports the important role of 4-HBA in microbial cross talk, as 4-HBA strongly inhibits hyphal formation by *Candida albicans*. Together, our findings describe the dual roles of 4-HBA in both intraspecies signaling and interkingdom communication.

**KEYWORDS** quorum sensing, *Shigella sonnei*, 4-hydroxybenzoic acid, physiology, virulence

**S**higella is a genus of Gram-negative bacteria and human intestinal pathogens, which lead to severe diarrhea and have become an important cause of mortality and morbidity worldwide (1, 2). *Shigella* belongs to the *Enterobacteriaceae* family and is usually transmitted through ingestion of contaminated food and water (3). Based on a combination of biochemical and serological characteristics, *Shigella* species are divided into the following subgroups: *S. dysenteriae*, *S. boydii*, *S. flexneri*, and *S. sonnei*. To date, the rate of *S. sonnei* infection is outpacing that of *S. flexneri* (4–6). Therefore, clarifying the virulence regulation mechanism of *S. sonnei* has become an important task for addressing global health concerns.

Quorum sensing (QS) is a process of cell-to-cell communication that is employed by Gram-positive and Gram-negative bacteria to regulate many biological functions as

Address correspondence to Yinyue Deng, dengyle@mail.sysu.edu.cn.

The authors declare no conflict of interest.

10.1128/spectrum.04835-22 1

well as interspecies and interkingdom communication (7–15). Recent studies have shown that bacteria can regulate their own physiological state and participate in intraspecies signaling and interkingdom communication by sensing anthranilic acid (16–18). In addition, several studies have demonstrated that diffusible factor 4-hydroxybenzoic acid (4-HBA) plays an important role in controlling the physiology and virulence of bacteria. It was reported that 4-HBA is synthesized by the bifunctional enzyme XanB2 in *Xanthomonas campestris* pv. *campestris* and is involved in metabolic processes and pathogenicity (19–21). Moreover, it was demonstrated that 4-HBA binds to the regulator protein $LysR_{Le}$ to enhance its activity to control the production of the antifungal metabolite heat-stable antifungal factor (HSAF) and promote the competitive advantage of bacteria (22).

Cross talk mediated by small molecules between different microorganisms is ubiquitous in microbial communities (23, 24). Many QS signals were identified to be involved in interspecies or interkingdom communication. It was reported that *cis*-2-dodecenoic acid (BDSF) biosynthesized by *Burkholderia* strains alters the expression of multiple virulence-related genes and inhibits biofilm formation in *Francisella novicida* Utah112 (25). BDSF also interferes with the morphological transition of *Candida albicans* (26, 27). Previous studies have shown that the 3-oxo-C12-homoserine lactone signal produced by *Pseudomonas aeruginosa* mediates a competitive relationship between *P. aeruginosa* and *C. albicans* (28, 29). Conversely, farnesol, the QS signal produced by *C. albicans*, significantly attenuated the virulence of *P. aeruginosa* by impacting synthesis of the *Pseudomonas* quinolone signal (PQS) (30). Moreover, the QS signal autoinducer-2 (AI-2) also participates in cross talk between different bacterial species (31–33).

The human gut microbiota involves a complicated ecosystem in which hundreds of bacterial species interact with each other (34). This knowledge inspired us to investigate the critical roles of QS signals in intestinal pathogenic bacteria in humans. In this study, we found that 4-HBA produced by UbiC modulates biofilm formation, EPS production, and pathogenicity in *S. sonnei*. The exogenous addition of 4-HBA restored these phenotypes of the *ubiC* mutant strain to the wild-type strain levels. Furthermore, 4-HBA bound to AaeR and enhanced the binding of AaeR to the promoters of target genes. Interestingly, we found that the 4-HBA signaling system controls the biosynthesis of anthranilic acid, which might play an important role in the physiology and pathogenicity of *S. sonnei*. Furthermore, 4-HBA inhibited hyphal formation and biofilm formation in *C. albicans* and effectively improved the competitive advantage of *S. sonnei* against *C. albicans*. Our data suggest that 4-HBA employed by *S. sonnei* plays important roles in both intraspecies signaling and cross talk in ecological niches.

## RESULTS

**The *S. sonnei* extract inhibits hyphal formation and biofilm formation by *C. albicans*.** *C. albicans* is an opportunistic pathogenic fungus that belongs to the human intestinal flora. Considering the high incidence of *Shigella* in intestinal diseases, we studied whether *Shigella* and *C. albicans* competed or cooperated with each other in the intestinal environment. Low-molecular-weight compounds were extracted from the liquid culture of *S. sonnei* using ethyl acetate, which was concentrated and quantified by rotary steaming and dissolved in methanol. The morphological transition of *C. albicans* is a significant virulence factor, as hyphal formation plays critical functions in the infection process. Therefore, we tested the efficacy of the extract and discovered that it significantly inhibited hyphal development (Fig. 1a and b). The results also demonstrated that the extract inhibited biofilm formation in *C. albicans* in a dose-dependent manner (Fig. 1c); however, the extract did not obviously affect the growth of *C. albicans* (see Fig. S1 in the supplemental material), revealing that *S. sonnei* may release small compounds that inhibit biofilm formation and morphological transition in *C. albicans*; hence, *S. sonnei* gains a competitive advantage in the human intestinal environment.

**The major active component of *S. sonnei* is 4-hydroxybenzoic acid.** To identify the active components of *S. sonnei* that inhibit biofilm formation and the morphological transition of *C. albicans*, supernatant from a 100-L *S. sonnei* culture medium was

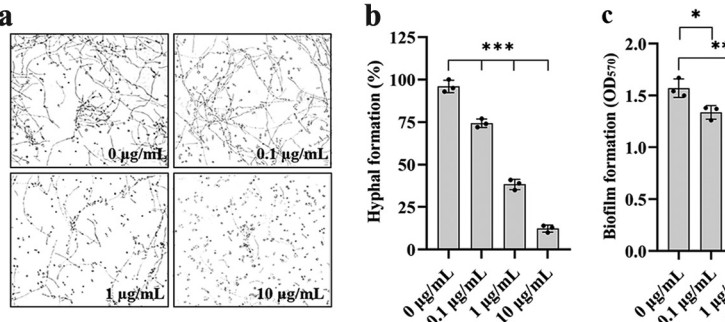

**FIG 1** Effects of the ethyl acetate extract of *S. sonnei* on *C. albicans*. (a to c) The extract of *S. sonnei* inhibited the hyphal growth (a and b) and biofilm formation (c) of *C. albicans*. The concentrations of the extract used were 0, 0.1, 1, and 10 $\mu$g/mL. The extract was dissolved in methanol, and the same volume of methanol (used as the solvent for the extract) was used as a control. *C. albicans* was grown in YNB medium containing 2% glucose and was treated with different concentrations of the extract. Samples were withdrawn after incubation at 37°C for 8 h and photographed at ×60 magnification. The hyphal formation of the wild-type strain was defined as 100% to normalize the hyphal formation ratios of the samples treated with different extract concentrations. The data are presented as the means $\pm$ SD of three independent experiments. Error bars indicate SDs. The significance of the results was determined by one-way ANOVA (*, $P < 0.05$; **, $P < 0.01$; ***, $P < 0.001$; ns, no significance).

isolated and purified by high-performance liquid chromatography (HPLC). Approximately 18.4 mg of purified compound, which showed inhibitory activity against the formation of hyphae by *C. albicans*, was obtained. The results of structural characterization showed four protons in the aromatic region in the $^1$H nuclear magnetic resonance (NMR) spectrum (Fig. 2a). The $^{13}$C NMR data of the compound showed the presence of six aromatic carbons and one carbonyl carbon (Fig. 2b, Table S1). Electrospray ionization-tandem mass spectrometry (ESI-MS) analysis of the active compound revealed a molecular ion $[M-H]^-$ with an *m/z* ratio of 137.0248 (Fig. 2c), matching the molecular formula of $C_7H_5O_3$. The results were consistent with the literature (35), which indicated that the active compound was 4-hydroxybenzoic acid (4-HBA) (Fig. 2d).

**4-HBA reduces hyphal formation by interfering with the cyclic AMP-dependent signaling pathway of *C. albicans*.** To determine whether the effects of 4-HBA on *C. albicans* are specific and related to the dose, different concentrations of 4-HBA were used, and the inhibitory effects on the morphological transition and biofilm formation were evaluated, which are related to the pathogenicity of *C. albicans*. The results showed that 4-HBA displayed a significant inhibitory effect on the formation of hyphae and biofilms by *C. albicans* in a dose-dependent manner (Fig. S2a to c). Moreover, we also measured the effect of 4-HBA on the growth rate of *C. albicans* and found that the exogenous addition of 200 $\mu$M 4-HBA showed little influence on the growth rate of *C. albicans* (Fig. S2d). We continued to test whether 4-HBA interfered with the signaling pathways involved in hyphal development and biofilm formation to precisely determine the inhibitory mechanism of 4-HBA on *C. albicans*. Mitogen-activated protein kinase (MAPK) and cyclic AMP-dependent pathways are two key signal transduction pathways associated with pathogenicity in *C. albicans* (36, 37). Quantitative reverse transcription-PCR (RT-qPCR) analysis showed that the exogenous addition of 4-HBA inhibited the expression levels of *Hwp1*, *Als1*, *Efg1*, *Ece1*, and *Tec1*, all of which belong to the cAMP-dependent pathways (Fig. S2e and f). Taken together, these results suggest that 4-HBA inhibited hyphal formation and biofilm formation by interfering with the cAMP-dependent signal pathways of *C. albicans*.

**UbiC is responsible for 4-HBA biosynthesis in *S. sonnei*.** In *Escherichia coli*, the chorismate pyruvate lyase encoded by *ubiC* is the key enzyme involved in the biosynthesis of 4-HBA (38, 39). To identify the genes responsible for 4-HBA biosynthesis, *ubiC* homologs were first searched in the genome sequence of *S. sonnei* Ss046 and identified as SSON_4219 using the National Center for Biotechnology Information Basic Local Alignment Search Tool (BLAST) program (Fig. 3a and b). In-frame deletion of *ubiC*

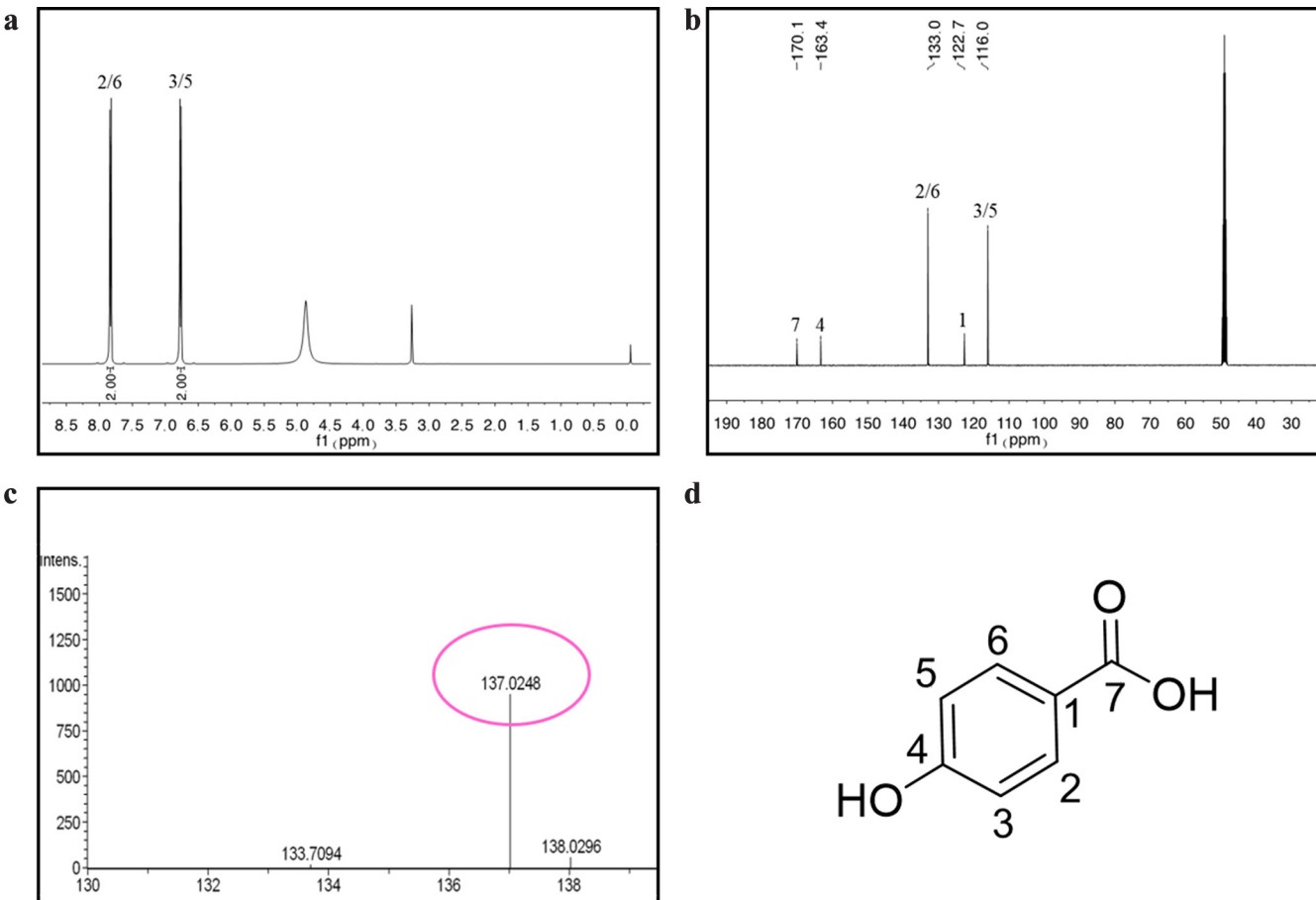

**FIG 2** Structural characterization of 4-HBA. (a) $^1$H NMR spectrum of 4-HBA. (b) $^{13}$C NMR spectrum of 4-HBA. (c) ESI-MS spectra of 4-HBA. (d) Chemical structure of 4-HBA.

completely abolished 4-HBA production in *S. sonnei* (Fig. 3c) but did not affect the growth rate of the bacterial cells in LB medium (Fig. S3). To further confirm the enzymatic activity observed *in vitro*, the fusion protein UbiC, which contains 166 amino acids and has a calculated molecular weight of 22.4 kDa, was purified using affinity chromatography (Fig. 3d). *In vitro* enzyme activity analysis showed that UbiC directly catalyzed the conversion of chorismic acid to 4-HBA (Fig. 3e, Fig. S4).

**Deletion of *ubiC* impairs biological functions in *S. sonnei*.** It was determined that 4-HBA inhibited hyphal formation and biofilm formation in *C. albicans*, and we continued to investigate whether this compound plays a role in regulating the physiology of *S. sonnei*. As in-frame deletion of *ubiC* completely abolished the production of 4-HBA, we then tested the phenotypes of biofilm formation and EPS production in the *ubiC* deletion mutant. The biofilm formation and EPS production by the *ubiC* mutant were decreased by 40% and 51%, respectively, compared to those of the wild-type strain (Fig. 3f and g). Interestingly, both overexpression of *ubiC* and exogenous addition of 4-HBA restored the phenotypes of the *ubiC* mutant strain to the wild-type strain levels. Moreover, compared to the *S. sonnei* wild-type strain, the cytotoxicity was attenuated by 41% when HeLa cells were incubated with the *ubiC* mutant strains at 8 h postinoculation (Fig. 3h). We also found that deletion of *ubiC* impaired the inhibition of *S. sonnei* on the hyphal formation and biofilm formation of *C. albicans* (Fig. S5), suggesting that 4-HBA mediated the interkingdom competition between *S. sonnei* and *C. albicans*.

**The production of 4-HBA is cell density dependent.** To investigate whether the biosynthesis of 4-HBA is related to cell density, we first measured the time course of 4-

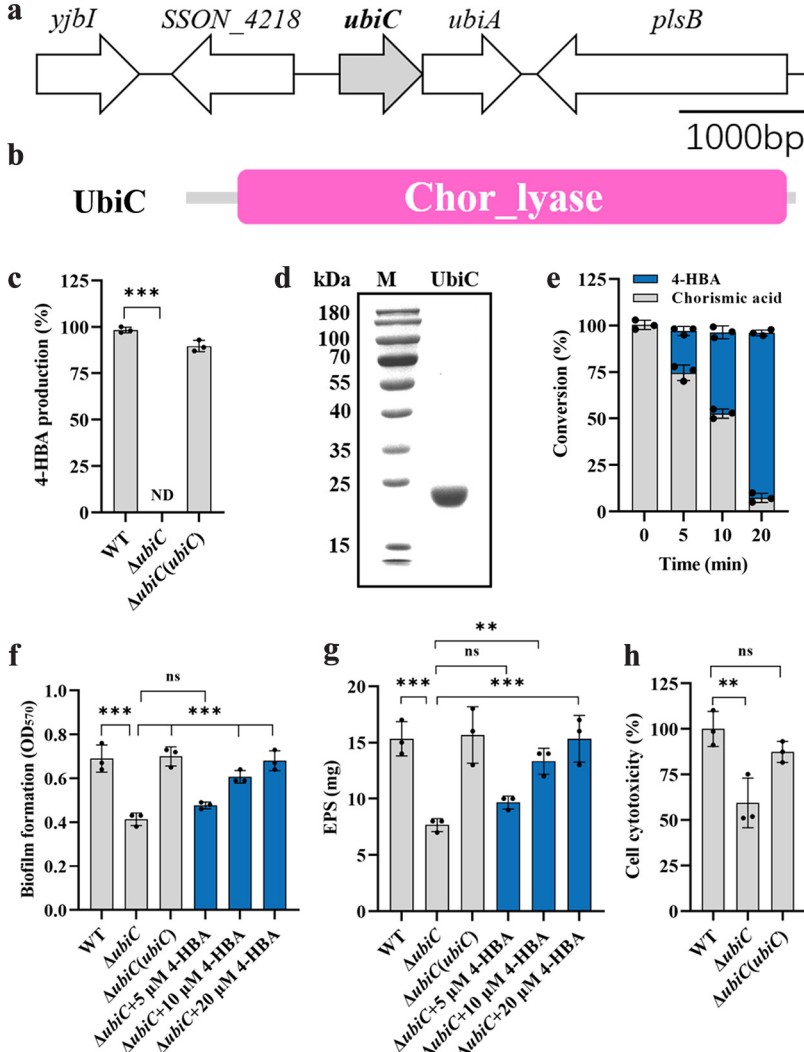

**FIG 3** Analysis of the biological functions of 4-HBA in *S. sonnei*. (a) Genomic organization of the *ubiC* region in *S. sonnei*, which was based on the genome in the model for the bacterium *S. sonnei* Ss046. (b) Domain structure analysis of the UbiC protein. (c) Detection of 4-HBA production via the LC-MS assay. For convenient comparison, 4-HBA production in the *S. sonnei* wild-type strain was arbitrarily defined as 100% and used to normalize the production ratios of other strains. (d) SDS-PAGE analysis of the UbiC protein. (e) Analysis of the production of 4-HBA by UbiC to catalyze chorismic acid transformation. (f and g) The virulence-related phenotypes of biofilm formation (f) and EPS production (g) in the *S. sonnei* wild-type, *ubiC* mutant, and *ubiC* complemented strains and *ubiC* mutant with the addition of different concentrations of 4-HBA were examined. (h) The cell cytotoxicity of *S. sonnei* wild-type, *ubiC* mutant, and *ubiC* complemented strains was evaluated by lactate dehydrogenase (LDH) assay. The value of the wild-type strain was defined as 100% to normalize the LDH release ratios of the samples treated with the other strains. The data are presented as the means ± SD of three independent experiments. Error bars indicate the SDs. The significance of the results was determined by one-way ANOVA or two-way ANOVA (*, $P < 0.05$; **, $P < 0.01$; ***, $P < 0.001$; ns, no significance). ND, not detected.

HBA production by determining 4-HBA concentrations at various growth stages. The yield of 4-HBA was low in the early stages of growth, increased dramatically when the cell density was high after 4 h, and peaked at 10 h, and then the 4-HBA concentration began to decline (Fig. 4a).

We continued to study the transcriptional profile of *ubiC*, which is essential for 4-HBA production. A 495-bp DNA sequence containing the *ubiC* promoter region was transcriptionally fused to the *luxCDABE* coding region and introduced into the wild-type and *ubiC* mutant strains. The transcriptional level of *ubiC* in the wild-type strain gradually increased, peaked at 10 h postinoculation, and then decreased (Fig. 4b), which was well correlated with the 4-HBA accumulation profile.

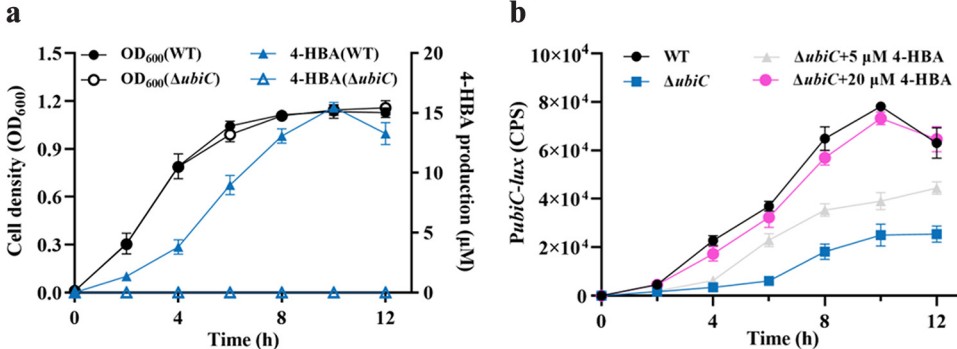

**FIG 4** Analysis of 4-HBA production and *ubiC* transcriptional expression. (a) Time-course analysis of 4-HBA accumulation and cell growth in the *S. sonnei* wild-type strain and the *ubiC* deletion mutant strain in liquid medium. (b) The gene expression levels of *ubiC* in the *S. sonnei* wild-type strain and the *ubiC* deletion mutant strain and the *ubiC* deletion mutant strain supplemented with 4-HBA. The gene expression levels of *ubiC* were evaluated by assessing the light production (counts per second [cps]) of the *ubiC-luxCDABE* transcriptional fusions in the *S. sonnei* strains. The data are presented as the means ± SD of three independent experiments. Error bars indicate the SDs.

To determine whether the transcriptional expression of *ubiC* is autoregulated by 4-HBA, we compared the transcriptional profiles of *ubiC* in the wild-type strain, the *ubiC* deletion mutant strain, and the *ubiC* deletion mutant strains with the addition of 4-HBA. It was observed that promoter activity by the mutant strain was decreased significantly from that of the wild-type strain. In addition, exogenous addition of 4-HBA at a final concentration of 20 $\mu$M almost restored the promoter activity of *ubiC* in the *ubiC* deletion mutant strain to the levels of the wild-type strain, suggesting that the production of 4-HBA may be autoregulated at the transcriptional level (Fig. 4b).

**4-HBA controls the expression levels of a wide range of genes.** To further investigate the regulatory roles of *ubiC* in controlling bacterial biological functions, the transcriptomes of the wild-type strain and the *ubiC* mutant strain were analyzed and compared using RNA sequencing (RNA-Seq). Differential gene expression analysis showed that 151 genes were upregulated and 42 genes were downregulated in the *ubiC* mutant compared with the wild-type strain (Fig. S6a). These differentially expressed genes were associated with a range of biological functions, including transport, translation, nitrate metabolism, flagellar self-assembly and oxidative phosphorylation (Fig. S6b). Interestingly, among these differentially expressed genes, the genes related to nitrate metabolism were significantly increased (Table S2), including *napA*, *napB*, *narK*, and *narJ*, which are all involved in the process of reducing nitrates to nitrites. In addition, compared with that of the wild-type strain, the transcriptional expression levels were downregulated for *ubiD* and *ubiX*, which are related to the 4-HBA metabolic pathway and responsible for catalyzing the decarboxylation of 3-octaprenyl-4-hydroxy benzoate to 2-octaprenylphenol. The results of RT-qPCR also determined the reliability of the RNA-Seq data (Fig. S6c).

**AaeR is a key downstream regulator of the 4-HBA signaling system.** To further determine the regulatory mechanism of 4-HBA, we continued to identify the downstream components of the 4-HBA signaling system in *S. sonnei*. It was previously reported that AaeR regulates the expression levels of efflux proteins by responding to 4-HBA, salicylate, benzoate, and 1-naphthoate at millimolar levels (40). We then hypothesized that AaeR might be involved in 4-HBA signaling in *S. sonnei*, searched for homologs of AaeR in the genome sequence of *S. sonnei* Ss046 and identified SSON_3385 using the BLAST program. We *trans* expressed SSON_3385 (AaeR) in the 4-HBA-deficient mutant *ubiC*, and the results showed that the *trans* expression of AaeR rescued the defective biofilm formation and EPS production by the *ubiC* mutant strain (Fig. 5a and b). These results inspired us to continue investigating the role of AaeR in regulating these biological functions. An in-frame deletion mutant of *aaeR* was generated, and we found that deletion of *aaeR* resulted in the same phenotypic changes as those observed with the 4-HBA-deficient

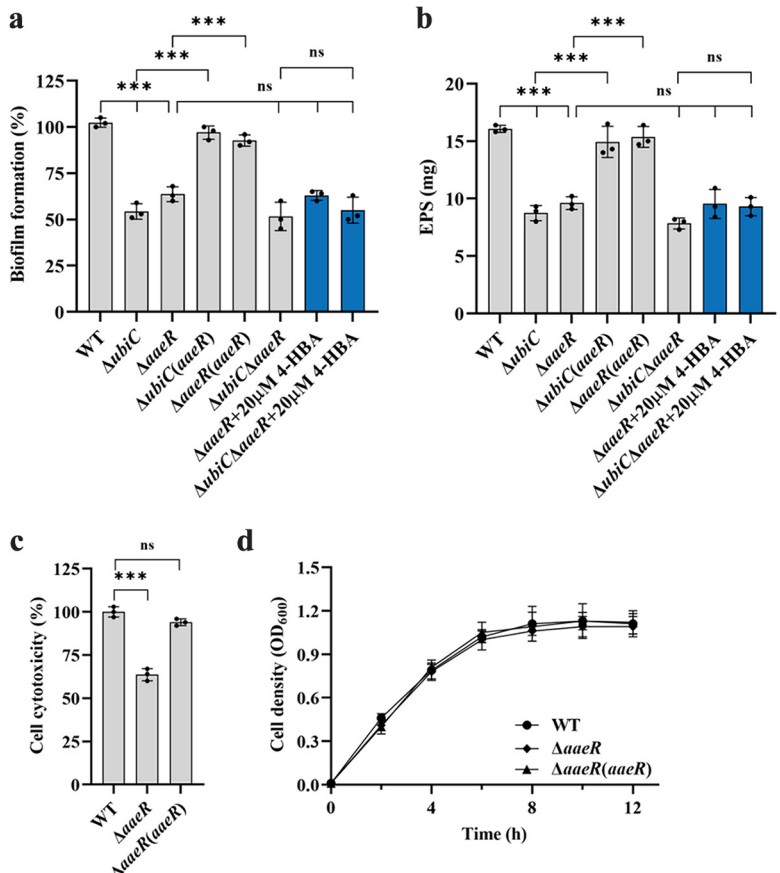

**FIG 5** Analysis of the biological functions of AaeR in *S. sonnei*. (a and c) The virulence-related phenotypes of biofilm formation (a) and EPS production (b) in the *S. sonnei* strains were examined. (c) The cell cytotoxicity of *S. sonnei* strains was evaluated by an LDH assay. (d) The growth curve of *S. sonnei* strains. For convenient comparison, the value of the wild-type strain was defined as 100% in panels a and c to normalize the ratios of the value from other strains. The data are presented as the means $\pm$ SD of three independent experiments. Error bars indicate the SDs. The significance of the results was determined by one-way ANOVA or two-way ANOVA (*, $P < 0.05$; **, $P < 0.01$; ***, $P < 0.001$; ns, no significance).

mutant *ubiC* (Fig. 5a and b). Adding exogenous 4-HBA did not restore the phenotypes of the *aaeR* deletion mutant or the *ubiC* and *aaeR* double deletion mutant to wild-type strain levels (Fig. 5a and b). In addition, the *aaeR* deletion mutant exhibited cytotoxicity in human HeLa cells that was decreased by 36% compared to that of the wild-type strain (Fig. 5c), but the bacterial cells showed a similar growth rate (Fig. 5d). From this result, we concluded that AaeR is a key downstream regulator of the 4-HBA signaling system in *S. sonnei*.

**4-HBA is a signaling ligand of AaeR.** It was determined that AaeR controls the same biological functions as the 4-HBA signal, and 4-HBA positively controls the expression of AaeR, as the transcription level of *aaeR* was downregulated in the deletion mutant *ubiC* (Fig. S6c and S7). Since AaeR contains a LysR_substrate domain, we hypothesized that AaeR may also modulate biological functions by interacting with 4-HBA (Fig. 6a). To confirm our hypothesis, we performed microscale thermophoresis (MST) assays to test whether AaeR could bind to 4-HBA. AaeR, which contained 304 amino acids (aa) with a calculated molecular weight of 33.7 kDa, was purified to homogeneity using affinity chromatography and characterized for interactions with 4-HBA (Fig. 6b). AaeR bound to 4-HBA with an estimated dissociation constant ($K_d$) of 23.05 $\pm$ 0.76 $\mu$M (Fig. 6c).

AaeR is a key downstream regulator of the 4-HBA signaling system and contains an HTH domain that is predicted to be closely related to DNA binding. Therefore, we tested whether the transcriptional regulation of target genes can be achieved through

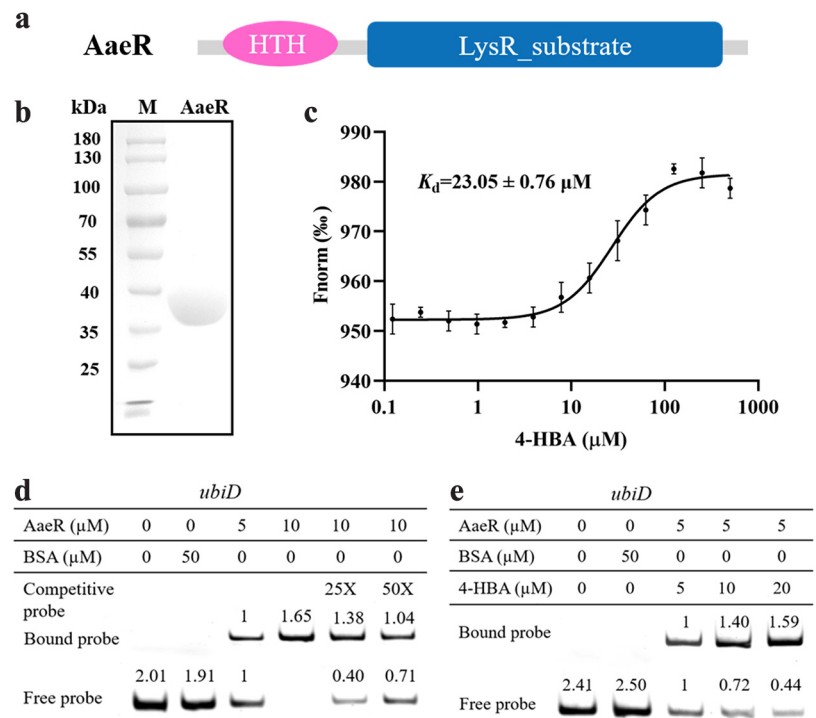

**FIG 6** Analysis of the regulatory mechanism of AaeR in *S. sonnei*. (a) Domain structure analysis of the AaeR protein. (b) SDS-PAGE of the AaeR protein. (c) The binding constant of 4-HBA and AaeR detected by MST. (d) EMSA analysis of the binding of AaeR to the *ubiD* promoters. (e) The effects of 4-HBA on the binding of AaeR to the promoters of *ubiD*. "Fnorm (‰)" indicates the fluorescence time trace changes in the MST response. Biotin-labeled 184-bp *ubiD* promoter DNA probes were used for the protein binding assay. A protein-DNA complex, which is represented by a band shift, was formed when AaeR protein was incubated with the probe at room temperature for 30 min. The band intensities were quantified and analyzed using the program ImageJ. The relative intensity of the corresponding band was compared with the levels in lane 3 set to 1. The data are presented as the means ± SD of three independent experiments. Error bars indicate SDs.

direct binding of AaeR to their promoters by performing electrophoretic mobility shift assays (EMSAs). UbiD was revealed to be involved in the metabolic process of 4-HBA (41), in which the transcriptional level was significantly downregulated in the *ubiC* mutant strain (Fig. S6c). We found that deletion of *aaeR* also caused a reduction in the expression of *ubiD* at the transcriptional level (Fig. S8), suggesting that *ubiD* is a target gene controlled by AaeR. Therefore, *ubiD* was chosen for further study. A PCR-amplified 184-bp DNA fragment from the *ubiD* promoter was used as a probe. The EMSAs confirmed the binding of AaeR to the *ubiD* promoter DNA fragments, and the amounts of probe bound to AaeR increased with increasing amounts of AaeR (Fig. 6d).

To determine how the binding of 4-HBA to AaeR might affect the activity of AaeR, we then tested the effects of 4-HBA on the binding of AaeR to the *ubiD* promoter by EMSA. As shown in Fig. 6e, the binding of AaeR to the *ubiD* promoter probes was enhanced when 4-HBA was present, and the binding activity of AaeR increased with increasing amounts of 4-HBA. Together, these results suggest that the addition of 4-HBA enhanced the activity of AaeR to control target gene expression.

**4-HBA signal affects anthranilic acid production and occurs in many bacteria.** Notably, the KEGG metabolic pathway showed that 4-HBA and anthranilic acid share the same precursor, chorismic acid (Fig. S9). We then investigated whether the metabolism of 4-HBA affected the production of anthranilic acid. The *trpED* transcriptional levels in the *ubiC* mutant strains were higher than those in the wild-type strains evaluated by the reporter system (Fig. 7a), which was constructed by transcriptionally fusing the *trpE* promoter to *luxCDABE*. The RT-qPCR results also showed that the transcription levels of *trpE* and *trpD*, which encode anthranilate synthase components

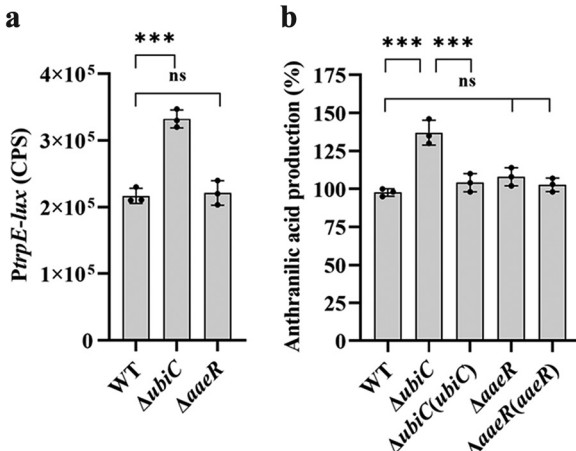

**FIG 7** Effects of *ubiC* on the biosynthesis of anthranilic acid. (a) Analysis of the expression levels of *trpE* in the wild-type strain, *ubiC* deletion mutant strain, and *aaeR* deletion mutant strain. The gene expression levels of *trpE* were evaluated by assessing the light production of the *trpE-luxCDABE* transcriptional fusions in the *S. sonnei* strains. (b) Detection of anthranilic acid production in the *S. sonnei* strains via the LC-MS assay. For convenient comparison, anthranilic acid production in the *S. sonnei* wild-type strain was arbitrarily defined as 100% and used to normalize the production ratios of other strains. The data are presented as the means ± SD of three independent experiments. Error bars indicate the SDs. The significance of the results was determined by one-way ANOVA or two-way ANOVA (*, $P < 0.05$; **, $P < 0.01$; ***, $P < 0.001$; ns, no significance).

annotated in KEGG, were upregulated in the *ubiC* mutant strain compared to the wild-type strain (Fig. S10). Consistently, the production of anthranilic acid in the *ubiC* mutant strain was higher than that in the wild-type strain, but the production of anthranilic acid was not affected in the *aaeR* mutant strain, suggesting that another unknown downstream component might be involved in the metabolism of anthranilic acid (Fig. 7b). These findings suggest that there is a connection between the 4-HBA signaling system and the metabolism of anthranilic acid, which is a signal in both *Pseudomonas aeruginosa* and *Ralstonia solanacearum* (16–18).

To investigate whether the 4-HBA signaling system is widely present in bacteria, both UbiC and AaeR homologs were searched for in the genome database by BLAST. It was indicated that the 4-HBA signaling system might be widely conserved in many different bacterial species, including *Acinetobacter*, *Citrobacter*, *Klebsiella*, *Vibrio*, and *Halopseudomonas* (Table S3 and S4).

## DISCUSSION

The human intestinal tract is inhabited by a multitude of microorganisms, which are collectively referred to as the gut microbiota (42, 43). *Shigella* species are important pathogens in human intestines that have been confirmed to be major contributors to diarrhea (1). Data have shown that the number of patients infected with *S. sonnei* has gradually surpassed those with *S. flexneri* in recent years, suggesting that *S. sonnei* exhibits more competitive advantages in the intestinal environment (5). Previous studies have shown that *S. sonnei* encodes a type VI secretion system (T6SS) that provides a competitive advantage in the gut, as *S. sonnei* competes against *E. coli* and *S. flexneri* in mixed cultures, and this advantage is reduced in T6SS mutant strains (44). In addition, the interkingdom communication of small molecules, which are represented by QS signals and enhance the competitive advantage of bacteria, has also attracted widespread attention (27, 45). In this study, we found that 4-HBA played an important role in microbial ecology. We demonstrated that 4-HBA from *S. sonnei* interferes with *C. albicans* hyphal growth and biofilm formation by inhibiting the cAMP-dependent signaling pathways of *C. albicans* (Fig. S2). Since *S. sonnei* and *C. albicans* are important human intestinal pathogens, our data showed that *S. sonnei* also employed 4-HBA to participate in interkingdom communication to improve its competitiveness.

Previous studies have shown that 4-HBA is vital to bacterial physiological activities, including the production of EPS, oxidative stress, synthesis of antibacterial compounds, and virulence (19–22). Our study demonstrated that 4-HBA not only affected the competitive ability of *C. albicans* but also regulated considerable biological functions in *S. sonnei*. The deletion of 4-HBA synthase reduced biofilm formation, EPS production and the cytotoxicity of *S. sonnei* (Fig. 3f to h). It was demonstrated that coenzyme Q8 (CoQ8) is an essential element for aerobic respiratory growth and is related to bacterial virulence, whose biosynthesis needs 4-HBA as the precursor. Our data showed that the deletion of 4-HBA synthase eliminated CoQ8 biosynthesis (Fig. S11). Transcriptome data showed that compared with that of the wild-type strain, the transcriptional levels of nearly 200 genes were significantly changed in the *ubiC* mutant strains. Interestingly, genes related to nitrate metabolism were significantly upregulated (Fig. S6c, Table S2). Nitrate reduction, also known as denitrification, mainly occurs in the process of anaerobic respiration. The results suggested that the metabolism of 4-HBA might affect the aerobic respiration of *S. sonnei*.

AaeR is a regulator that belongs to the LysR transcriptional regulatory family and contains a substrate binding region and a classic DNA binding region HTH; this protein is also thought to be a virulence regulator that is activated by QS and is involved in activating the expression of locus of enterocyte effacement (LEE) pathogenicity genes in enterohemorrhagic and enteropathogenic *E. coli* (46). Previous studies revealed that AaeR responds to several aromatic carboxylic acid compounds to control the transcriptional expression of efflux pump genes and hence restore cellular homeostasis in *E. coli* (40). Su et al. demonstrated that a LysR-family transcription factor present in *Lysobacter enzymogenes* (LysR$_{Le}$; the homologous protein of AaeR) bound with 4-HBA and enhanced its ability to form complexes with DNA (22). We also found a homologous protein of AaeR in *S. sonnei* and verified its ability to bind to 4-HBA by MST. The phenotypes of *aaeR* mutant strains were consistent with those of the *ubiC* deletion mutant strains, and the exogenous addition of 4-HBA did not rescue the defective phenotypes to the levels of the wild-type strain, suggesting that AaeR played an important role in the 4-HBA regulatory network (Fig. 5a and b). Intriguingly, the EMSA showed that AaeR bound to the promoter of *ubiD* and accelerated the metabolism of 4-HBA (Fig. 6d and e) but did not bind to the promoter of *ubiC* (Fig. S12a and b), the expression of which was controlled by 4-HBA (Fig. 4b). Furthermore, the transcription level of *ubiC* was not affected in the *aaeR* mutant strain compared to the wild-type strain (Fig. S13). One more piece of evidence is that the production of anthranilic acid was only affected in the *ubiC* mutant strain but was not changed in the *aaeR* mutant strain (Fig. 7b), suggesting that there might be another unknown receptor of 4-HBA in *S. sonnei*.

In addition, both UbiC and AaeR homologs were searched in bacteria using BLAST, and the results revealed that both 4-HBA synthase and the receptor were highly conserved in many other bacteria. Intriguingly, some bacteria present only one homolog of UbiC or AaeR, suggesting that 4-HBA functions as an intraspecies signal or a signal for cross talk between different microorganisms. With these data together, we demonstrated that the 4-HBA signaling system is widely distributed in bacteria, and further investigating the roles and mechanisms of this signaling system in other bacteria is of great significance.

## MATERIALS AND METHODS

**Strains, culture, and agents.** The bacterial strains used in this study are listed in Table S5. All bacterial strains and plasmids used in this study have been sequenced. *S. sonnei* and *E. coli* strains were cultured in Luria-Bertani (LB) medium (10 g/L tryptone, 5 g/L yeast extract, 10 g/L NaCl; pH 7.4) or LB agar (LB medium containing 15 g/L agar) at 37°C. The *Candida* strain used in this study was *C. albicans* SC5314 (ATCC MYA-2876), which was grown in either 6.7 g/L yeast nitrogen broth lacking amino acids (YNB) supplemented with 2% glucose or YPD medium (1% yeast extract, 2% peptone, and 2% dextrose) at 30°C. The antibiotics were added to the medium according to the experimental needs, and the following antibiotics were used in this work: chloramphenicol (50 $\mu$g/mL), kanamycin (100 $\mu$g/mL), and ampicillin (100 $\mu$g/mL). 4-HBA (GC content, ≥98%) was purchased from Solarbio and dissolved in dimethyl sulfoxide (DMSO) to a final concentration of 100 mM for stock.

**Purification and structural analysis of 4-HBA.** The purification and characterization of the active fractions followed the same procedure as previously described (47). Briefly, *S. sonnei* cells were cultured overnight in LB medium, and the pH of the supernatant was adjusted to 4 with hydrochloric acid and then extracted with an equal volume of ethyl acetate. After the extract was evaporated and concentrated, the residue was quantified and dissolved in methanol. The extract was analyzed by HPLC using a $C_{18}$ reverse-phase column (Atlantis T3 column, 5 $\mu$m, 4.6 mm by 250 mm) and eluted with methanol-water (from 5:95 to 70:30 vol/vol) at a flow rate of 1 mL/min. Finally, the active fractions were detected, concentrated, and further purified by HPLC using a semipreparative $C_{18}$ reverse-phase column. Peaks were monitored using a UV detector at 210 nm and 254 nm.

The $^1$H and $^{13}$C nuclear magnetic resonance (NMR) spectra were recorded on an AVANCE III HD 400 (temperature, 298.0 K; Bruker, Billerica, MA, USA) operating at 400 MHz for $^1$H or 101 MHz for $^{13}$C. Ultra-high-performance liquid chromatography (UHPLC)-ESI-MS/MS was performed in an LC-30A UHPLC system (Shimadzu Corporation, Kyoto, Japan) with a Waters $C_{18}$ column (1.8 $\mu$m, 150 by 2.1 mm) and a Shimadzu 8060 QQQ-MS mass spectrometer with an ESI source interface.

**Construction of the *S. sonnei* mutant and complemented strains.** All primer sequences are summarized in Table S6 in the supplemental material. The *ubiC* and *aaeR* mutants were generated using the $\lambda$ Red recombinase system, and this was supported by the pKD46 plasmid encoding three proteins (Exo, Beta, and Gam) necessary for homologous recombination. The chloramphenicol-resistant pKD3 plasmid was used as the template, and primers were designed to cover the 39-bp homologous arm sequences. The pCP20 plasmid was used to remove chloramphenicol resistance. In addition, the target gene was integrated into the plasmid to obtain the complemented strains by using pUC18. The resulting constructs were introduced into *S. sonnei* deletion mutants using electroporation.

**Construction of reporter strains.** Plasmid pMS402 carrying a promoterless *luxCDABE* reporter gene cluster was used to construct promoter-*luxCDABE* reporter fusions as reported previously (48, 49). The target promoter was amplified by PCR and then cloned upstream of the *lux* genes on pMS402 with the BamHI-XhoI site. The resultant plasmid was transformed into *S. sonnei* by electroporation. Promoter activities were measured as light production (counts per second [cps]) using an Infinite E PLEX Microchor Board detector.

**Biofilm formation assays.** The *S. sonnei* biofilm assays were set up in 96-well plates and quantified by crystal violet staining as previously described (50). Briefly, overnight cultures were diluted to an optical density at 600 nm ($OD_{600}$) of 0.05 in the suspension with LB broth. The inoculated plates were incubated for 24 h at 37°C without shaking. The wells were gently washed with phosphate-buffered saline (PBS) three times after removing the medium. The air-dried samples were fixed with methanol and then stained with 100 $\mu$L of 0.5% crystal violet solution for 15 min at room temperature. The wells were washed with distilled water 3 times to completely remove the crystal violet. Finally, 150 $\mu$L of 95% ethanol was added to the wells to dissolve the samples. For *C. albicans*, the strains were cultured in YNB medium supplemented with 2% glucose and incubated for 8 h at 37°C without shaking. Biofilm formation was quantified by reading the microplates at 570 nm using a Multiskan Spectrum device.

**EPS content detection.** The *S. sonnei* suspensions ($OD_{600}$, 0.05) were processed with the same method as the biofilm formation assay. All samples were inoculated for 24 h at 37°C with shaking, and then the samples ($OD_{600}$, 3.0) were centrifuged at 12,000 $\times$ *g* for 30 min to remove the precipitate. The supernatant was treated with 4 volumes of absolute ethanol at 4°C for 12 h and then centrifuged at 4°C and 12,000 $\times$ *g* for 30 min. The supernatant was removed, and the samples were air-dried at 55°C. The mass of the precipitate was calculated, and the experiment was repeated three times.

**Cell cytotoxicity assays.** Cytotoxicity assays were performed according to previously described methods (47). In brief, HeLa cells were grown in Dulbecco's modified Eagle's medium (DMEM) supplemented with 10% fetal bovine serum (FBS) in 96-well tissue culture plates at $1 \times 10^5$ cells/well. The bacterial cells were cultured in LB medium at 37°C overnight, centrifuged, and resuspended in DMEM (1% FBS). HeLa cells were infected with bacterial cells at $10^9$ CFU/mL for 8 h. The amount of lactate dehydrogenase (LDH) released was determined with a CytoTox 96 kit (Promega, Wisconsin, USA). The results of the cytotoxicity assay were quantified by measuring the absorbance at 490 nm, and the cytotoxicity was calculated relative to that of an uninfected control.

**Filamentation assays.** The filamentation assays followed the same procedure as previously described (51). *C. albicans* cells were cultured overnight and then inoculated in fresh YNB medium supplemented with 2% glucose to an $OD_{600}$ of 0.1 to induce hyphal growth. Different concentrations of compounds were added at the final concentrations indicated, and then the *C. albicans* cells were incubated for 8 h at 37°C. Images of cells were photographed at $\times$60 magnification.

**Growth rate analysis.** Bacterial strains were cultured in LB medium overnight, and then the cultures were washed twice in fresh LB medium and inoculated into fresh media to an $OD_{600}$ of 0.05. Growth curves were constructed in triplicate by incubating the cells for 12 h at 37°C with shaking at 220 rpm. For *C. albicans*, the strains were cultured in YNB medium supplemented with 2% glucose and incubated at 30°C for 24 h with shaking at 220 rpm. The growth was determined by measuring the optical density at 600 nm.

**RNA-Seq and RT-qPCR.** RNA-Seq and quantitative real-time PCR (RT-qPCR) validation were performed. Two groups, the wild-type *S. sonnei* and the $\Delta ubiC$ mutant, were used for these analyses. Briefly, a total amount of 3 $\mu$g RNA per sample was used as input material for the RNA sample preparations. Sequencing libraries were generated using the NEBNext Ultra RNA library prep kit for Illumina (New England Biolabs [NEB], USA) following the manufacturer's recommendations, and index codes were added to assign sequences to each sample. The library preparations were sequenced on an Illumina HiSeq 2000 platform. For RNA-Seq data analysis, the index of the reference genome (GenBank accession

number CP053751) was built using Bowtie2 v2.2.6, and paired-end clean reads were aligned to the reference genome using Bowtie2. Differential expression analysis of two conditions/groups (two biological replicates per condition) was performed using the DESeq2 R package (v1.16.1). A $q$ value (false-discovery rate) of <0.05 and fold change of >1 were used to identify the significantly differentially expressed genes (DEGs). DEGs were visualized by a volcano plot and used for gene ontology (GO) and Kyoto Encyclopedia of Genes and Genomes (KEGG) functional enrichment analyses to identify which DEGs were significantly enriched in GO terms or metabolic pathways. Furthermore, a StepOne real-time PCR system (ABI, Foster City, CA) was employed to validate the RNA-Seq results by detecting the expression levels of DEGs. The gene names and primer sequences used for RT qPCR are shown in Table S6. The *hisG* of *S. sonnei* was used as the internal control.

**Protein expression and purification.** The intact *ubiC* and *aaeR* genes were cloned into the pET28a (+) vector and then transformed into competent *E. coli* BL21(DE3) cells. Protein expression was performed in LB medium supplemented with 100 $\mu$g/mL kanamycin. Cells were grown at 37°C and 220 rpm to and OD$_{600}$ of about 0.4 to 0.6. Recombinant gene expression was induced overnight at 16°C by the addition of 1 mM IPTG (isopropyl-$\beta$-D-thiogalactopyranoside). After induction, the cells were harvested by centrifugation at 4,000 $\times$ $g$ for 20 min. The cells were resuspended in PBS (pH 7.4) and disrupted by sonication. The resultant suspension was centrifuged for 15 min at 4°C and 10,000 $\times$ $g$, and the supernatant was loaded onto a 1-mL His-Tag column (GE Healthcare), which was previously equilibrated with 10 bed volumes of PBS containing 250 mM NaCl. The recombinant proteins were eluted with a 10 to 300 mM imidazole linear gradient in PBS buffer. Finally, protein purification was evaluated by SDS-PAGE.

**Quantitative analysis of 4-HBA and anthranilic acid.** Bacteria were freshly inoculated and cultured overnight at 37°C with shaking, and then the bacterial liquid was diluted to an OD$_{600}$ of 0.05 in fresh medium and cultured overnight until the OD$_{600}$ reached 3.0. The supernatant of the bacterial liquid was collected by centrifugation and extracted with the same amount of ethyl acetate. The extract was evaporated, concentrated, and then dissolved in methanol. Finally, the extract was subjected to a Shimadzu LC-30A UHPLC system for quantitative analysis.

**Quantitative analysis of CoQ8.** CoQ8 was extracted as previously described (20). Briefly, bacteria were freshly inoculated and cultured overnight at 37°C with shaking, and then the bacterial liquid was diluted to an OD$_{600}$ of 0.05 in fresh medium and cultured overnight until the OD$_{600}$ reached 3.0. The cells were collected by centrifugation and resuspended in 0.2 M acetate buffer (pH 5.6) after being washed twice with PBS buffer. The cell homogenate was sonicated for 120 s, and a hexane-acetone (1:1, vol/vol) reaction mixture was added, followed by sonication and vortexing. The hexane fraction was evaporated, and the residue was then dissolved in chloroform-methanol (1:1, vol/vol), followed by washing with 0.7% NaCl. The chloroform fraction was evaporated, concentrated, and then dissolved in methanol. Finally, the extract was subjected to a Shimadzu LCMS-8060 for quantitative analysis.

**Microscale thermophoresis assay.** Protein-binding experiments were carried out with a Nano Temper 16 Monolith NT.115 instrument (NanoTemper Technologies; www.nanotemper-technologies.com). In brief, AaeR protein was labeled with the L014 Monolith NT.115 protein labeling kit (NanoTemper, Munich, Germany). Labeled AaeR protein and 4-HBA were mixed and loaded onto standard treated silicon capillaries (K022 Monolith NT.115, NanoTemper, Munich, Germany), and fluorescence was measured. The measurements were carried out at 60% light-emitting diode (LED) power and 40% MST power.

**Electrophoretic mobility shift assay.** EMSA was performed according to the instructions for the Thermo Fisher Scientific kit with minor modifications. In brief, the 3-ends of the promoters purified by PCR were labeled with biotin using Thermo Fisher Scientific's Biotin 3'-end DNA labeling kit. Protein-DNA binding interactions were detected using a light shift chemiluminescent EMSA kit. DNA-protein binding reactions were performed according to the manufacturer's instructions (Thermo Fisher, Waltham, MA, USA). A 5% polyacrylamide gel was used to separate the DNA-protein complexes. After UV cross-linking, the biotin-labeled probes were detected in the membrane using a biotin luminescence detection kit (Thermo Fisher).

**Statistical analysis.** Statistical analyses were performed using Prism 8 software (GraphPad). The data are presented as the means $\pm$ standard deviations (SD) of three independent experiments. The unpaired $t$ test between two groups, one-way analysis of variance, or two-way analysis among multiple groups was used to calculate $P$ values. Statistical significance is indicated as follows: ns, no significance; *, $P < 0.05$; **, $P < 0.01$; ***, $P < 0.001$. All results were calculated from the average of at least three replicates.

**Data availability.** Data supporting the findings of this study are available within the paper and its supplemental material.

## SUPPLEMENTAL MATERIAL

Supplemental material is available online only.

**SUPPLEMENTAL FILE 1**, PDF file, 1.7 MB.

## ACKNOWLEDGMENTS

This work was financially supported by the National Key Research and Development Program of China (2021YFA0717003) and the Science, Technology, and Innovation Commission of Shenzhen Municipality (no. JCYJ20200109142416497).

M.W. and Y.D. designed the research; M.W., J.Z., Y.Z., X.C., Q.G., H.T., and B.C. performed the research; M.W., S.S., and Y.D. analyzed the data; and M.W. and Y.D. wrote the paper.

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
