## [Reviewer comments · Microbiology Spectrum]

Microbiology Spectrum

A 4-hydroxybenzoic acid-mediated signaling system controls the physiology and virulence of *Shigella sonnei*

Mingfang Wang, Jia Zeng, Yu Zhu, Xiayu Chen, Quan Guo, Huihui Tan, Binbin Cui, Shihao Song, and Yinyue Deng

Corresponding Author(s): Yinyue Deng, Sun Yat-Sen University

Review Timeline:

Submission Date:	November 24, 2022
Editorial Decision:	January 27, 2023
Revision Received:	February 20, 2023
Accepted:	March 15, 2023

Editor: Guoliang Qian

Reviewer(s): The reviewers have opted to remain anonymous.

Transaction Report:

DOI: <https://doi.org/10.1128/spectrum.04835-22>

January 27, 2023

Prof. Yinyue Deng
Sun Yat-Sen University
Shenzhen
China

Re: Spectrum04835-22 (A 4-hydroxybenzoic acid-mediated signaling system controls the physiology and virulence of *Shigella sonnei*)

Dear Prof. Yinyue Deng:

Link Not Available

Sincerely,

Guoliang Qian

Journals Department
Reviewer comments:

Reviewer #1 (Comments for the Author):

This work demonstrated that 4-HBA act as a quorum sensing signaling molecule in *Shigella sonnei* by presenting the following evidence: 1. the production of 4-HBA in *Shigella sonnei* is positively correlated with the bacterial cell density; 2. the UbiC enzyme is responsible for synthesizing 4-HBA and expression of UbiC is auto-regulated by 4-HBA; 3. 4-HBA is recognized by the transcription regulator AaeR, enhances AaeR-DNA interaction, and influences the expression of a wide range of genes. Moreover, the authors also showed 4-HBA is a potential inter-kingdom signaling molecule interfering with morphological transition of *Candida albicans*. Although 4-HBA as a QS signal has been investigated in several bacterial strains, this work highlighted the dual roles of 4-HBA in both intraspecies signaling and interkingdom communication. Overall, this work is interesting and technically sound.

Major concerns:

1. Besides its potential role as a QS signal, 4-HBA has been long known as the precursor of Q8, which is an essential element for aerobic respiratory growth. Deletion of UbiC should completely eliminate Q8 biosynthesis and have wide metabolic consequences (attenuated virulence for example) and this could be an alternative way to explain the current data and should be extrapolated in the discussion.
2. The expression of UbiC is auto-regulated by 4-HBA, but 4-HBA does not activate AaeR nor AaeR directly regulates UbiC. In this sense, 4-HBA and AaeR are not analogous to the classical QS signal and receptor. The authors may comment on this point.
3. I'm surprised the RNA-seq data showed 4-HBA decreased the expression of ubiD, which was later showed to be potentially positively regulated by 4-HBA (EMSA data).

Minor points:

1. Line 230: should be "4-HBA is a signaling ligand of AaeR. "
2. Line 316: should be "...to the promoter of ubiC (Fig. S11a, b), the expression of which was controlled by 4-HBA"

Reviewer #2 (Comments for the Author):

In the manuscript of Wang et al. " A 4-HBA mediated signaling system controls the physiology and virulence of *Shigella sonnei*", the authors found that extract of *S. sonnei* culture inhibits biofilm formation and development of *C. albicans*. The revealed that the active chemical is 4-HBA. In addition, the also found that 4-HBA binds a transcription regulator AaeR to enhance the activity of this TF in controlling downstream expression. This work is well designed and the experimental data is enough to support the conclusion, the results are interesting in understanding the bacteria-fungi interaction and interkingdom communication. I have several minor concerns on the work:

1. The authors found the mixture of *S. sonnei* extract inhibits biofilm formation and development of *C. albicans* (Fig. 1) and revealed that 4-HBA is a major bioactive compound (Fig. S2). However, did 4-HBA the only active component in inhibition or there are other unidentified chemicals in the extract? How many percentages of 4-HBA in the mixture?
2. Lines 142-148, the authors checked the transcription level of MAPK marker genes and conclude the change of MAPK gene expressions is the reason to result in biofilm formation of fungi, this is not comprehensive because they did not check the other pathways.
3. Fig. 6d, in the EMSA experiment, there is only a 50 times competition, please added more competitions with various concentrations of un-labelled probe. In addition, please quantify the density of the bands

Staff Comments:

Preparing Revision Guidelines

Please return the manuscript within 60 days; if you cannot complete the modification within this time period, please contact me. If you do not wish to modify the manuscript and prefer to submit it to another journal, please notify me of your decision immediately so that the manuscript may be formally withdrawn from consideration by Microbiology Spectrum.

If your manuscript is accepted for publication, you will be contacted separately about payment when the proofs are issued;

please follow the instructions in that e-mail. Arrangements for payment must be made before your article is published. For a complete list of **Publication Fees**, including supplemental material costs, please visit our website.

Point-to-point response to reviewers' suggestions for Spectrum04835-22

Reviewer comments:

Reviewer #1 (Comments for the Author):

This work demonstrated that 4-HBA act as a quorum sensing signaling molecule in *Shigella sonnei* by presenting the following evidence: 1. the production of 4-HBA in *Shigella sonnei* is positively correlated with the bacterial cell density; 2. the UbiC enzyme is responsible for synthesizing 4-HBA and expression of UbiC is auto-regulated by 4-HBA; 3. 4-HBA is recognized by the transcription regulator AaeR, enhances AaeR-DNA interaction, and influences the expression of a wide range of genes. Moreover, the authors also showed 4-HBA is a potential inter-kingdom signaling molecule interfering with morphological transition of *Candida albicans*. Although 4-HBA as a QS signal has been investigated in several bacterial stains, this work highlighted the dual roles of 4-HBA in both intraspecies signaling and interkingdom communication. Overall, this work is interesting and technically sound.

Major concerns:

1. Besides its potential role as a QS signal, 4-HBA has been long known as the precursor of Q8, which is an essential element for aerobic respiratory growth. Deletion of UbiC should completely eliminate Q8 biosynthesis and have wide metabolic consequences (attenuated virulence for example) and this could be an alternative way to explain the current data and should be extrapolated in the discussion.

Response: Thanks for your good suggestion, we have added this experiment result as Supplementary Figure 11 as suggested. Our result showed that the deletion of *ubiC* completely abolished the production of CoQ8 (Line 293-297).

2. The expression of UbiC is auto-regulated by 4-HBA, but 4-HBA does not activate AaeR nor AaeR directly regulates UbiC. In this sense, 4-HBA and AaeR are not analogous to the classical QS signal and receptor. The authors may comment on this point.

Response: Thanks for your good suggestion. The expression of UbiC is auto-regulated by 4-HBA, but AaeR does not regulate UbiC. This result indicates that 4-HBA may have another potential receptor, which regulates the expression level of UbiC by responding to 4-HBA. Similarly, the expression level of TrpE and TrpD are affected by UbiC, but AaeR does not regulate them. Together, it suggests that there may be another potential receptor

of 4-HBA that regulates the expression of UbiC, TrpD and TrpE. We have added this comment in the discussion (Line 322).

3. I'm surprised the RNA-seq data showed 4-HBA decreased the expression of *ubiD*, which was later showed to be potentially positively regulated by 4-HBA (EMSA data).

Response: Thanks for your suggestion. The RNA-seq data were analyzed and compared by using the *ubiC* mutant strain and the wild-type strain, we could find that deletion of *ubiC* resulted in the reduction of *ubiD* expression, suggesting that 4-HBA positively regulates the expression of *ubiD*.

Minor points:

1. Line 230: should be "4-HBA is a signaling ligand of AaeR. "

Response: Thanks for your suggestion, we have revised it as suggested.

2. Line 316: should be "...to the promoter of *ubiC* (Fig. S11a, b), the expression of which was controlled by 4-HBA"

Response: Thanks for your suggestion, we have revised it as suggested.

Reviewer #2 (Comments for the Author):

In the manuscript of Wang et al. " A 4-HBA mediated signaling system controls the physiology and virulence of *Shigella sonnei*", the authors found that extract of *S. sonnei* culture inhibits biofilm formation and development of *C. albicans*. The revealed that the active chemical is 4-HBA. In addition, the also found that 4-HBA binds a transcription regulator AaeR to enhance the activity of this TF in controlling downstream expression. This work is well designed and the experimental data is enough to support the conclusion, the results are interesting in understanding the bacteria-fungi interaction and interkingdom communication. I have several minor concerns on the work:

1. The authors found the mixture of *S. sonnei* extract inhibits biofilm formation and development of *C. albicans* (Fig. 1) and revealed that 4-HBA is a major bioactive compound (Fig. S2). However, did 4-HBA the only active component in inhibition or there are other unidentified chemicals in the extract? How many percentages of 4-HBA in the mixture?

Response: Thanks for your suggestion. Our data suggested that 4-HBA is not the only active component in inhibition of biofilm formation and hyphae development of *C. albicans* (data unpublished). The dry weight of the extract was approximately 10.8 g, and approximately 18.4 mg of 4-HBA was obtained, therefore, 4-HBA is about 0.17% in the mixture of *S. sonnei* extract.

2. Lines 142-148, the authors checked the transcription level of MAPK marker genes and conclude the change of MAPK gene expressions is the reason to result in biofilm formation of fungi, this is not comprehensive because they did not check the other pathways.

Response: Thanks for your suggestion. In fact, we had detected two signaling pathways,

MAPK and cAMP, which are two key signaling pathways associating with biofilm formation and hyphae formation in *C. albicans*. We have detected a total of 15 related genes, including *Hwp1*, *Cek1*, *Als3*, *Efg1*, *Cst20*, *Als1*, *Pde2*, *Hst7*, *Tec1*, *Cdc35*, *Cph1*, *Ece1*, *Hsp90*, *Rim101* and *Ras1*. The results showed that the exogenous addition of 4-HBA inhibited the expression levels of *Hwp1*, *Als1*, *Efg1*, *Ece1*, and *Tec1*, all of which belong to the cAMP-dependent pathways (Fig. S2e, f). So, our conclusion is that 4-HBA inhibited hyphal formation and biofilm formation mainly by interfering with the cAMP-dependent signal pathways of *C. albicans*.

3. Fig. 6d, in the EMSA experiment, there is only a 50 times competition, please added more competitions with various concentrations of un-labelled probe. In addition, please quantify the density of the bands

Response: Thanks for your suggestion, we have revised the figures as suggested.

ubiD						
AaeR (μM)	0	0	5	10	10	10
BSA (μM)	0	50	0	0	0	0
Competitive probe					25X	50X
Bound probe			1	1.65	1.38	1.04
Free probe	2.01	1.91	1		0.40	0.71

ubiD					
AaeR (μM)	0	0	5	5	5
BSA (μM)	0	50	0	0	0
4-HBA (μM)	0	0	5	10	20
Bound probe			1	1.40	1.59
Free probe	2.41	2.50	1	0.72	0.44

March 15, 2023

Prof. Yinyue Deng
Sun Yat-Sen University
Guangzhou 510642
China

Re: Spectrum04835-22R1 (A 4-hydroxybenzoic acid-mediated signaling system controls the physiology and virulence of *Shigella sonnei*)

Dear Prof. Yinyue Deng:

Your manuscript has been accepted, and I am forwarding it to the ASM Journals Department for publication. You will be notified when your proofs are ready to be viewed.

Sincerely,

Guoliang Qian
Editor, Microbiology Spectrum
